# SONGCOMPOSER: A LARGE LANGUAGE MODEL FOR LYRIC AND MELODY COMPOSITION IN SONG GENERATION

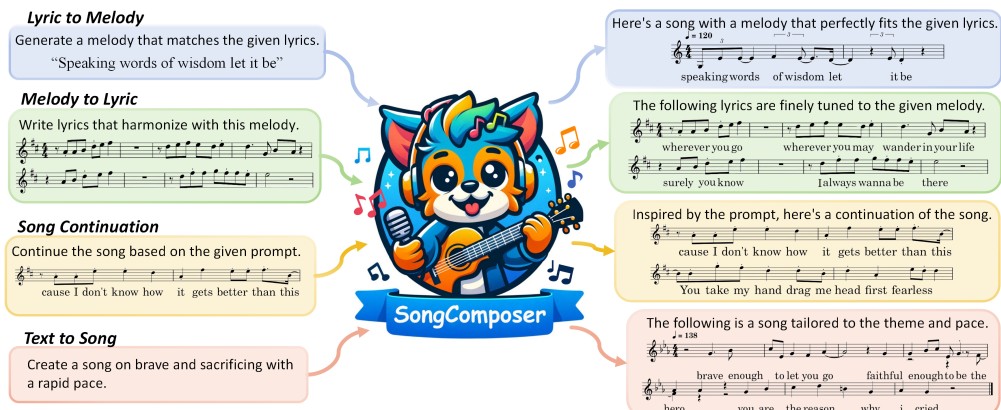

Figure 1: Overview of the song-related instruction-following composition by SongComposer. Song-Composer utilizes symbolic song representation to compose melodies tailored to lyrics, craft lyrics to complement melodies, extend existing songs, and generate new songs from textual prompts.

## ABSTRACT

A song typically comprises the vocal track and the music track. Creating lyrics and melodies for the vocal track in a symbolic format, known as song composition, plays a significant role in the song generation. This delicate and complex task demands expert musical knowledge of melody, an advanced understanding of lyrics, and precise alignment between them. Despite achievements in sub-tasks such as lyric generation, lyric-to-melody, and melody-to-lyric, etc, a unified model for song composition has not yet been achieved. In this paper, we introduce SongComposer, **a pioneering step towards a unified song composition model that can readily create symbolic lyrics and melodies following instructions.** SongComposer is a music-specialized large language model (LLM) that, for the first time, integrates the capability of simultaneously composing lyrics and melodies into LLMs. To achieve this goal, three non-trivial efforts are introduced. **1) Sheet music understanding**, we designed a flexible tuple format to load lyric and note attributes, fostering word-level alignment between lyrics and melodies, and enabling SongComposer to generate lyrics with accompanying well-aligned melodies. **2) Song note tokenizing**, the vocabulary of the tokenizer is extended for song notes, and we find a proper scalar-manner initialization of new tokens based on musical prior is essential for the model to understand musical rhythm. **3) Structural music generation**, we propose a multi-stage pipeline for progressively capturing the musical structure. Initially, we extract and feed motif-level melody patterns to SongComposer to build its basic generation capabilities. Later, we insert special tokens into the whole-song data to denote phrase-level structure, promoting logical repetition and smooth coherence. Extensive experiments demonstrate that SongComposer outperforms advanced LLMs, including GPT-4, in tasks such as lyric-to-melody generation, melody-to-lyric generation, song continuation, and

text-to-song creation. We showcase the generated samples on our anonymous project page[1]. Due to the lack of high-quality symbolic song datasets with lyrics and melodies, we have carefully curated and will publicly release SongCompose, a large-scale song pretraining and supervised finetuning dataset that includes lyrics, melodies, and paired lyrics-melodies in both Chinese and English.

# 1 INTRODUCTION

Symbolic song composition aims to generate the vocal track of a song as a sequence of symbols representing lyrics and melodies. It is a vital task in song generation and requires professional knowledge. Recently, this field has become a highly active area of research in both academic and industrial domains. Previous efforts have made significant progress in isolated sub-tasks of song composition such as lyric generation (Zhang et al., 2022c), lyric-to-melody (Yu et al., 2021a; Ju et al., 2022; Sheng et al., 2021; Zhang et al., 2022a) or melody-to-lyric generation (Sheng et al., 2021; Ma et al., 2021). However, the absence of a unified framework for generating both lyrics and melodies concurrently while adhering to specific instructions poses a challenge for seamless adaptation, thereby creating a higher hurdle for everyday amateurs.

The recent surge in large language models (LLMs) has dramatically revolutionized the artificial intelligence landscape, especially in natural language understanding and generation (Brown et al., 2020; Chiang et al., 2023; Wei et al., 2021; Chowdhery et al., 2023; Raffel et al., 2020; Devlin et al., 2018). These models have established new benchmarks for parsing and producing human language, showcasing human-level proficiency in complex language environments. Given that symbolic song representation shares structural similarities with human language, it seems plausible that LLMs could facilitate the creation of symbolic songs. Furthermore, unlike previous non-LLM methods (Sheng et al., 2021; Ju et al., 2022) that handle only specific tasks, LLMs can integrate various sub-tasks of song composition into a single model due to their instruction-following capabilities.

However, enabling LLMs to compose full-length songs that harmonize melody and lyrics is not a trivial task. First, as illustrated in Figure 2(a), symbolic song representation would decompose a song into its lyrics and note attributes (pitch, beat) and form a strict word-level alignment. Therefore, aligning lyric and melody attributes in a unified and efficient manner for LLMs is indispensable yet remains unexplored. Secondly, a song typically features a well-organized and hierarchical structure (Dai et al., 2022). For example, a composer usually uses the concept of motif and phrase to enrich the unity of a song. A motif is a recurring musical idea that serves as a fundamental building unit, and a phrase is a broader segment of music that forms a complete thought or expression. As shown in Figure 2(b), a single song may have a clear high-level phrase structure like Verse-Chorus, and across the whole song, there may be repetitive patterns known as motifs. Thus, enhancing LLMs to understand these succinct musical structures is of vital importance and may require explicitly curated knowledge input and design. Last but not least, current symbolic song datasets (Yu et al., 2021b; Wang et al., 2022; Huang et al., 2021) are either limited in quantity or lacking in quality. They often miss precise alignments between melody and lyrics, impeding progress in symbolic song generation.

To address the aforementioned challenges, we introduce SongComposer, an LLM capable of generating whole-song compositions that harmoniously integrate both melodies and lyrics. To the best of our knowledge, this is the first attempt to generate lyrics and melody simultaneously using LLMs.

Specifically, we propose a word-level tuple format to construct melody and lyric attributes in a flexible and unified manner, providing an efficient interface for aligning melody and lyrics. Besides, we introduce a scalar initialization method to seamlessly initialize pitch tokens based on the existing vocabulary of LLMs. This method initializes a central pitch first and then sets the remaining note pitches as multiples of the central pitch embeddings. In this way, we explicitly introduce and reinforce the relationship between pitches to LLMs.

To learn the hierarchical structure of a song, we use a progressive training approach with Song-Composer, enabling the model to recognize patterns of motifs and phrases. Initially, we extract highly repetitive melody snippets and treat these as general motifs for motif-level melody training.

---

[1] https://songcomposer.github.io/

| Lyric | jingle | all | the | way | - |
|---|---|---|---|---|---|
| Note | B4 | D5 | G4 | A4 | B4 | rest |
| Pitch | 71 | 74 | 67 | 69 | 71 | - |
| Duration | 0.512 | 0.498 | 0.520 | 0.484 | 1.119 | 1.122 |
| Beat | 1 | 1 | 1 | 1 | 2 | 2 |

(a) Example of symbolic song representation.          (b) Illustration of hierarchical structure in a song.

Figure 2: (a) Symbolic song representation involves precise alignment of notes and lyrics; (b) The structure of a song often comprises motif-level and phrase-level concepts.

Subsequently, we insert special tokens to denote phrase concepts when training on the full-length song data, instructing the model to directly identify which parts of the song correspond to verses, choruses, or other phrases. Based on these designs, our model is encouraged to generate structure-aware compositions that exhibit motif-level and phrase-level coherence.

Regarding the dataset, we have carefully compiled and curated a comprehensive high-quality dataset, SongCompose. This dataset comprises 280K songs with pure lyrics, 20K sets of pure melodies, and 8K paired lyrics and melodies in both Chinese and English. Moreover, it covers not only the pretraining dataset but also the supervised fine-tuning dataset for LLMs. Notably, the paired data feature precise word-level alignment, and this portion has been curated from scratch. We believe this large-scale dataset can serve as a critical resource for training large language models, and we plan to release it to propel further research in this field.

We evaluate SongComposer on four song-related tasks, as shown in Figure 1. Extensive experiments demonstrate that SongComposer outperforms advanced GPT-4 and several open-source LLMs both in terms of quality and adherence to the prompt. Moreover, we excel in the traditional model (Sheng et al., 2021; Ju et al., 2022) on specific lyric-to-melody tasks. In addition, we conduct a thorough ablation study to verify the effectiveness of the proposed components. We also include a memorization test (Carlini et al., 2022; Agostinelli et al., 2023) to check for inappropriate copying from the dataset, revealing that SongComposer's output significantly differs from the original sequences in the pretraining dataset.

In short, our contributions are as follows:

- We introduce SongComposer, an LLM capable of generating whole-song singable sheets that include both melodies and lyrics with well-structured formats following instructions.
- We propose a novel scalar initialization for note pitches and integrate motif- and phrase-level knowledge to enhance the model's understanding of pitch attributes and song structure.
- We curate SongCompose, a high-quality pretraining and supervised fine-tuning dataset with 280K lyrics, 20K melodies, and 8K precisely aligned lyric-melody pairs in Chinese and English.
- Extensive experiments show SongComposer outperforms traditional composition models and advanced LLMs like GPT-4 in various song-related generation tasks.

## 2 RELATED WORK

**Symbolic Song Composition.** Symbolic song composition encompasses several key tasks, including the creation of song lyrics, the composition of melodies, and the mutual generation between lyrics and melodies. Lyric generation focuses on producing meaningful and coherent song lyrics using deep learning techniques(Malmi et al., 2016; Zhang et al., 2022c; Xue et al., 2021). The goal of melody generation (Wu et al., 2019; Colombo et al., 2017) is to autonomously create musical melodies that can stand alone. Taking a step further, lyric-to-melody generation (Yu et al., 2021a; Ju et al., 2022; Sheng et al., 2021; Zhang et al., 2022a) involves generating melodies that align with given

lyrics. The reverse task, melody-to-lyrics generation (Bao et al., 2019; Li et al., 2020; Sheng et al., 2021; Ma et al., 2021) focuses on producing lyrics that match a given melody. While these methods are effective within their specific song composition tasks, they often cannot handle comprehensive composition tasks with a single model. However, SongComposer can process both melody and lyrics simultaneously in a unified format with the power of LLMs.

**Symbolic Music Generation.** Recent years have seen significant progress in symbolic music generation. The majority of studies have focused on converting music information into symbolic-style tokens and then processing these sequences with Transformers (Qu et al., 2024; Huang et al., 2019; Huang & Yang, 2020; Lu et al., 2023; Yuan et al., 2024; Liang et al., 2024; Deng et al., 2024). Music Transformer (Huang et al., 2019) is a pioneering Transformer model that generates music with long-term structure by leveraging a novel memory-efficient relative attention mechanism. Building on Music Transformer, REMI (Huang & Yang, 2020), a novel MIDI-derived event representation, enhances models with beat-based awareness and improves rhythmic structure in the generation of expressive Pop piano compositions.

**Large Language Models.** Recent advancements in large language models (Raffel et al., 2020; Radford et al., 2018; Chowdhery et al., 2023; Touvron et al., 2023; OpenAI, 2023; Ouyang et al., 2022; OpenAI, 2022; Ouyang et al., 2022; Chiang et al., 2023; Qian et al., 2024) have significantly enhanced natural language processing, showcasing impressive capabilities across diverse tasks. In the domain of symbolic music creation, recent endeavors (Yuan et al., 2024; Deng et al., 2024) propose employing large language models for generating symbolic pure music. However, crafting compositions encompassing both lyrics and melodies with LLMs remains an open problem. Inspired by the powerful human-level language capabilities of LLMs, we have developed the first unified LLMs framework that expands their application to lyric and melody composition for song generation.

**Paired Lyric-Melody Singing Dataset.** Singing data annotated with paired lyrics and melodies is important for song generation. Specifically, JVS-MuSiC (Tamaru et al., 2020), PopCS (Liu et al., 2022), and OpenSinger (Huang et al., 2021) offer a broad range of singing data but lack the crucial lyric-melody temporal alignment. NUS-48E (Duan et al., 2013), NHSS (Sharma et al., 2021), Tohoku Kiritan (Ogawa & Morise, 2021), and Opencpop (Wang et al., 2022) provide singing corpora across English, Japanese, and Chinese with manually aligned lyrics and melodies. However, they are limited in scale, featuring only monophonic singers and styles. More recently, M4Singer (Zhang et al., 2022b) compiles approximately 700 Chinese songs with lyric and melody pairs, but this amount is still insufficient for training an LLM for symbolic music generation. In this work, we collect around 8K symbolic songs for both English and Chinese from scratch to train SongComposer.

## 3 SONGCOMPOSER

### 3.1 SYMBOLIC REPRESENTATION FOR LLMS

**Pure Melody Format.** Inspired by the beat-based REMI representation (Huang & Yang, 2020), we first decompose the notes into three symbolic attributes: note pitch $p$, note duration $d$, and rest duration $r$. The pitch range $p$ is from MIDI note numbers 48 to 83, corresponding to notes C3 to B5, which is the most common range for human vocal performance. Given the tempo of the melody, measured in beats per minute (bpm), we measure the note duration $d \in \mathbb{Z}$ and rest duration $r \in \mathbb{Z}$ in the number of 1/16 beat:

$$d_k = \phi(\frac{\text{bpm}}{60}(\text{note-end}_k - \text{note-start}_k) \times 16), \quad r_k = \phi(\frac{\text{bpm}}{60}(\text{note-start}_{k+1} - \text{note-end}_k) \times 16),$$

where note-start and note-end are times in seconds, $k$ denotes the note index number and $\phi(\cdot)$ is an operator that constrains the value to the nearest integer within the range $[1, 256]$.

Then each note of pure melody is formatted in a tuple as follows:

$$\langle \text{bom} \rangle \text{ bpm is } \{bpm\}. \text{ Total } \{num\} \text{ lines.}$$
$$\text{The 1-st line: } \langle p_1 \rangle, d_1 \mid \langle \text{rest} \rangle, r_1 \mid \langle p_2 \rangle, d_2 \mid \langle \text{rest} \rangle, r_2 \cdots$$
$$\text{The 2-nd line: } \cdots \langle \text{eom} \rangle$$

where we treat $\langle \text{rest} \rangle$ as a type of note and skip the rest tuple if $r < 8$. $\langle \text{bom} \rangle$ and $\langle \text{eom} \rangle$ indicate the beginning and end of the melody, respectively. Note that $\langle \cdot \rangle$ represents special tokens we add outside the existing vocabulary.

**Pure Lyric Format.** The lyrics share the same language as LLMs, thus it can be directly used without additional design. The input of the pure lyric is formatted as follows:

$\langle$bol$\rangle$ Chinese/English song. Total $\{num\}$ lines.

The 1-st line: $w_1\, w_2\, \cdots$

The 2-nd line: $\cdots$ $\langle$eol$\rangle$

where special tokens $\langle$bol$\rangle$ and $\langle$eol$\rangle$ indicate the beginning and the end of pure lyrics, respectively; $w$ denotes a word in the lyrics.

**Paired Data Format.** After defining the pure melody and lyrics separately, we explore methods to combine them into paired formats. To effectively integrate lyrics and melodies into SongComposer, we investigate three alignment methods at different granularities: song-level, line-level, and word-level. As depicted in Table 4, the experiment results demonstrate that the finest word-level alignment achieves the highest generation quality and alignment precision. This finding aligns with expectations, as word-level alignment provides a format similar to pure melody data and allows a more nuanced understanding of the relationship between lyrics and melody. Formally, the input of the word-level paired melody is formatted as follows:

$\langle$bop$\rangle$ Mandarin/English song. bpm is $\{bpm\}$. Total $\{num\}$ lines.

The 1-st line: $\langle p_1 \rangle, d_1, w_1 \,|\, \langle$rest$\rangle, r_1 \,|\, \langle p_2 \rangle, d_2, w_2 \,|\, \langle$rest$\rangle, r_2 \cdots$

The 2-nd line: $\cdots$ $\langle$eop$\rangle$

where special tokens $\langle$bop$\rangle$ and $\langle$eop$\rangle$ indicate the beginning and the end of pair data. When a single lyric word is sung to multiple musical notes, we add a numerical suffix to the word to specify which note the word corresponds to. We show the examples of each proposed format in Appendix D.

## 3.2 Pitch Initialization

Motivated by the strong logical and mathematical relationship between different pitches, we argue that initializing pitch tokens with a strong prior on their relationships would be beneficial for the model to interpret pitch elements. Therefore, we attempt four initialization methods for pitch tokens to verify our intuition.

**Average Initialization** creates the embedding for new pitch tokens $\langle p \rangle$ by averaging the existing token embeddings of left bracket ($\langle$), pitch number ($p$), and right bracket ($\rangle$).

**Gaussian Initialization** generates embeddings for new pitch tokens using a Gaussian distribution, with the mean and variance calculated from existing token embeddings.

**Interpolation Initialization** initializes the embeddings for the lowest and highest pitch tokens ($\langle 48 \rangle$ and $\langle 83 \rangle$) using Gaussian initialization. The embeddings for the pitches in between are linearly interpolated between these two.

**Scalar Initialization** begins by initializing a central pitch token $\langle 66 \rangle$ using Gaussian initialization. The embeddings for the remaining pitches are then set as multiples of this central pitch embedding, where multipliers range from $[-\ln(e + 17), \cdots, -\ln(e), \ln(e), \cdots, \ln(e + 17)]$. Compared to interpolation initialization, scalar initialization is more like a special form of extrapolation.

To further interpret the learned pitch tokens under different initialization methods, we visualize the embeddings in Figure 3. The average initialization distinguishes between pitch and rest tokens but fails to capture the inherent pitch information, resulting in a collapsed cluster. The Gaussian method fails to differentiate between pitch tokens and other tokens effectively and does not learn a discernible pattern. For the remaining two methods, both initialization methods result in distinct patterns. The interpolation method positions pitch tokens far away from other tokens, while the scalar method results in a pattern where the mean cluster still lies among the existing tokens. Therefore, scalar initialization stays closer to the existing semantic spaces which may lead to a better generation than the interpolation method.

Empirically, we find that the scalar initialization works best for pitch modeling. For more details, please refer to the ablation study as shown in Table 5. Therefore, we use scalar initialization on pitch tokens for SongComposer.

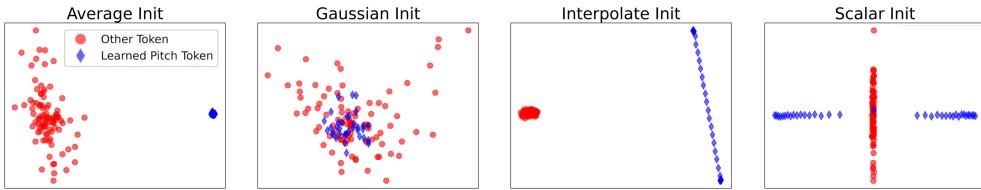

Figure 3: The visualization of learned pitch tokens and other tokens with different initialization methods. We use Principal Component Analysis (PCA) to reduce the dimensionality of the embeddings to 2 dimensions.

### 3.3 PROGRESSIVE STRUCTURE-AWARE TRAINING

Structure is crucial in song composition, with a typical song comprising multiple levels of structure (Dai et al., 2022). Therefore, we meticulously devise three stages of training for SongComposer to emphasize structural information at varying levels of time granularity.

**Motif-Level Melody Training.** Motif, in song composition, denotes a recurring musical idea that is key to enhancing the structure and coherence of the piece. Typically, a motif comprises a sequence of notes that repetitively appear throughout the song. Motivated by this concept, we intentionally select highly repetitive short note sequences to construct motif-level melody data. Subsequently, we kick off the training process of SongComposer by introducing this finely repetitive structure. In this way, the model is introduced to learn the motif-level composition.

**Independent Whole-Song Lyric and Melody Training.** After gaining insight into the basic units of composition through motif-level melody training, we extend the training of SongComposer to the whole-song level. However, directly training the model to establish alignments between melody and lyrics may expose challenges to SongComposer. Therefore, we continue to train the model using pure lyric and pure melody datasets to establish a foundation for basic whole-song understanding.

**Paired Lyric and Melody with Phrase-level Token Training.** Having a broader temporal dimension than a motif, the concept of phrases is also pivotal in structuring a song. A phrase is a sentence-level pattern that expresses a complete musical thought. To incorporate this understanding into composition, SongComposer trains on paired lyric and melody data, integrating the concept of phrases into the paired data. In our paper, we focus on five commonly used phrases such as 'intro', 'verse', 'chorus', 'bridge', and 'outro', while unifying less common phrases as 'other'. Each phrase in a song would be outlined by two special tokens to indicate its beginning and end, resulting in a total of $6 \times 2$ special tokens. To maintain the model's ability to process melodies and lyrics separately, we train an equal amount of pure melody, pure lyric, and paired data. In contrast to the previous stage, both the pure melody and pure lyric data are now decorated with phrase-level special tokens.

## 4 EXPERIMENTS

### 4.1 SONGCOMPOSE DATASET

To train the SongComposer, we curate a large-scale song pretraining and supervised fine-tuning dataset SongCompose. For more details, please refer to our Appendix A.

**Pure-lyric Dataset.** We collect 283K song lyrics from two online sources, including 150K English lyrics and 133K Chinese lyrics. After a series of lyric-cleaning processes, we gather high-quality lyrics from various genres and styles.

**Pure-melody Dataset.** We collect 20K MIDI files and extract melody attributes, including note pitch, note duration, and rest duration. We employ the *pretty_midi* Python module (Raffel & Ellis, 2014) to parse MIDI files and extract the "melody" or "vocal" tracks as the pure melody.

**Paired Lyric-melody Dataset.** We create from scratch a dataset of 8K pairs of lyrics and melodies from the Internet, with roughly half being in Chinese and the other half in English. Melodies and lyrics are matched at the word level.

**Supervised Finetuning Dataset.** We curate instruction-following data for song-generation tasks including creating melodies for given lyrics, writing lyrics for melodies, extending song segments, and generating songs from text descriptions. Specifically, we manually prepare 3K QA pairs for each of the first three tasks. Additionally, for the final task, we use GPT-4 to produce 1K song descriptions, which forms a text-to-song dataset that guides the song creation process.

## 4.2 Training Details

We adopt InternLM2-7B (Cai et al., 2024) as our base model and set the maximum token length as 5120. Except for the pitch tokens using Scalar initialization, all the other newly added special tokens adopt Gaussian initialization. We train the whole model to predict the next token based on prior text, maximizing the log-likelihood of tokens in the given examples. For optimization, we use AdamW optimizer (Loshchilov & Hutter, 2019) with a learning rate of $10^{-5}$, $\beta_1 = 0.9$, $\beta_2 = 0.95$, and a weight decay of 0.1. The entire dataset is iterated through once, with a batch size of 1. Additionally, a linear warm-up of the learning rate is applied during the initial $1\%$ of training steps, increasing from $10^{-6}$ to $10^{-5}$. Afterwards, a cosine schedule is applied, reducing the learning rate to a minimum of 0. This setting is consistent across both the pretraining and supervised fine-tuning stages. The whole training of SongComposer is conducted on 16 Nvidia A100 (80G) GPUs for approximately 2 days.

## 4.3 Objective Evaluation Metrics

We construct a validation set of 100 songs, evenly split between Chinese and English, none of which were seen by our model during training.

**Melody Generation.** For assessing the similarity between the generated melodies and the ground-truth, we adopt the metrics proposed by SongMASS (Sheng et al., 2021): Pitch Distribution Similarity (PD), Duration Distribution Similarity (DD), and Melody Distance (MD). Besides, we propose a recall rate to assess the repetition capability and partly indicate the structure within the song. This rate is calculated by dividing the total number of melodic lines by the number of unique melodic lines, with a minimum recall rate of 1 indicating no repetition.

**Lyric Generation.** We evaluate the similarity between generated and original lyrics using three metrics from different perspectives. We use a CoSENT (Cosine Sentence) model (Xu, 2023), specifically the base-multilingual version, to compute sentence-level cosine similarity. Additionally, we apply the ROUGE-2 score (Lin, 2004) to measure bigram overlap and the BERT score (BS) (Zhang et al., 2019) to assess similarity based on the contextual embeddings from the BERT-base model.

## 4.4 Subjective Evaluation Metrics

For the subjective evaluation, we conduct a user study with 30 participants 10 cases per task. We develop two metrics for each task and ask the participants to rate them. The rating scale is 1 to 5, where higher scores denote superior quality. In this way, we collect feedback on the quality of the generated content from a human perspective.

The evaluation criteria for different tasks are as follows: For Lyric-to-Melody Generation, we assess Harmony and Melody-Lyric Compatibility. For Melody-to-Lyric Generation, we evaluate Fluency and Melody-Lyric Compatibility. Song Continuation quality is assessed based on Overall Quality and Coherence to the Song Prompt. Text-to-Song generation is evaluated in terms of Overall Quality and Relevance to the Text Input.

In summary, each task is evaluated using two metrics: one that assesses the overall musical quality of the samples produced, and another that specifically addresses the challenges of each task. More detailed descriptions of the tasks and metrics can be found in Appendix B.

## 4.5 Experimental Results

We test and compare our method majorly with existing LLMs. For the alternative LLM baselines, we employ a few-shot prompt approach, feeding sample examples to prompt the LLM and produce the desired output following the given instructions. Details are provided in the Appendix E. We gather

Table 1: Objective evaluation of Lyrics-to-Melody and Melody-to-Lyrics tasks. For open-source LLMs, we select models with a size of 7B parameters. As for GPT models, we utilize the most recent versions, namely gpt-4-turbo and gpt-3.5-turbo. InternLM 2 + FT stands for fine-tuning the InternLM 2 without incorporating any proposed techniques in this paper.

| Method | Lyric-to-Melody | | | Melody-to-Lyric | | |
|---|---|---|---|---|---|---|
| | PD(%) ↑ | DD(%) ↑ | MD ↓ | Cosine Dist. ↑ | ROUGE-2 ↑ | BS ↑ |
| SongMass (Sheng et al., 2021) | 30.34 | 48.98 | 2.95 | 0.568 | 0.204 | 0.532 |
| TeleMelody (Ju et al., 2022) | 46.81 | 51.77 | 2.60 | - | - | - |
| LLaMA 2 (Touvron et al., 2023) | 12.10 | 32.56 | 9.21 | 0.625 | 0.153 | 0.560 |
| InternLM 2 (Team, 2023) | 16.32 | 34.25 | 5.77 | 0.636 | 0.124 | 0.505 |
| Qwen 1.5 (Bai et al., 2023) | 20.69 | 39.37 | 4.11 | 0.592 | 0.136 | 0.589 |
| GPT-3.5 (OpenAI, 2022) | 31.24 | 38.52 | 3.01 | 0.641 | 0.142 | 0.603 |
| GPT-4 (OpenAI, 2023) | 36.43 | 42.94 | 2.87 | 0.654 | 0.158 | 0.610 |
| InternLM 2 + FT | 24.78 | 38.96 | 4.03 | 0.621 | 0.144 | 0.575 |
| SongComposer | **50.75** | **57.71** | **2.20** | **0.697** | **0.234** | **0.657** |

Table 2: Subjective evaluation of four tasks. Harmony (HMY.), Melody-Lyric Compatibility (MLC.), Fluency (FLN.), Overall Quality (OVL.), Coherence to Song Prompt (COH.), and Relevance to Text Input (REL.) depict the quality of each method in generating musically harmonious, lyrically coherent, and contextually relevant songs.

| Method | Lyric-to-Melody | | Melody-to-Lyric | | Song Continuation | | Text-to-Song | |
|---|---|---|---|---|---|---|---|---|
| | HMY.↑ | MLC.↑ | FLN.↑ | MLC.↑ | OVL.↑ | COH.↑ | OVL.↑ | REL.↑ |
| GPT-3.5 (OpenAI, 2022) | 1.68 | 1.88 | 2.90 | 2.99 | 2.67 | 2.84 | 2.53 | 2.95 |
| GPT-4 (OpenAI, 2023) | 2.82 | 2.79 | 2.84 | 3.20 | 2.86 | 3.10 | 2.43 | 3.27 |
| SongComposer | **3.82** | **3.76** | **3.63** | **3.69** | **3.61** | **3.58** | **3.41** | **3.88** |

outputs from both GPT-4 and GPT-3.5 via their APIs. Additionally, we also assess other typical LLMs whose weights have been obtained from the Hugging Face community.

**Objective Evaluation.** Table 1 presents a comparison of methods for converting lyrics to melody and vice versa. Compared to traditional methods, which include special designs for specific tasks, Song-Composer still shows significant improvement. SongComposer significantly outperforms advanced large language models such as GPT-4 in both the lyric-to-melody task and the melody-to-lyrics task. Moreover, simple fine-tuning on InternLM 2 does not produce rational melodies and lyrics, showing the effectiveness of our systematic design. As shown in Table 3, SongComposer excels not only in generating high-quality lyrics and melodies individually but also in jointly producing both by continuing given lines. Since there is no objective evaluation for text-to-song generation, we showcase the results at this anonymous link [2] and provide a formatted musical score in Appendix F. The song generated by SongComposer is well-structured and coherent to the prompt.

**Subjective Evaluation.** The subjective evaluation in Table 2 highlights that SongComposer significantly surpasses GPT-3.5 and GPT-4 in overall quality, coherence to the prompt, and melody-lyric compatibility. This underscores SongComposer's advanced capability to capture the song's structure and generate a harmonized melody and lyrics that seamlessly fit together.

## 4.6 ABLATION STUDY

In the ablation study, we probe the SongComposer on song continuation task and report the Melody Distance (**MD**), Recall Rate (**RR**), and BERT Score (**BS**) to depict the quality of melody and lyric respectively. All studies are conducted on the validation set, except for the memorization analysis, where we use the training data to test whether the model memorizes the training set.

**Pair Alignment at Different Granularity.** We explore three methods for integrating lyrics and melody into a cohesive format. First, the **song-level** approach concatenates the entire set of lyrics for a song and the complete melody for that song. Second, the **line-level** method connects each line of

---

[2]https://songcomposer.github.io/

lyrics with the corresponding line of melody. Finally, the **word-level** method merges each individual word of the lyrics with a single note of the melody. We provide an example of the word-level pairing format in Section 3.1 and illustrate the other two alignment methods in Appendix C.1.

Table 3: Objective evaluation of song continuation. † means the exclusion of phrase-level tokens.

| Method | MD ↓ | BS ↑ | RR ↑ |
|---|---|---|---|
| GPT-3.5 (OpenAI, 2022) | 2.88 | 0.601 | 1.13 |
| GPT-4 (OpenAI, 2023) | 2.73 | 0.613 | 1.25 |
| SongComposer† | 2.58 | 0.612 | 1.35 |
| SongComposer | **2.12** | **0.662** | **1.64** |

Table 4: Ablation on alignment on lyric and melody at different granularities.

| Alignment | MD ↓ | BS ↑ |
|---|---|---|
| Song-Level | 3.71 | 0.572 |
| Line-Level | 2.42 | 0.623 |
| Word-Level | 2.12 | 0.662 |

Table 4 shows that finer alignment improves generation quality. Word-level alignment has the lowest melody distance and highest BERT score, indicating the best performance. Furthermore, we observe that the song-level and line-level pairing formats often fail to accurately produce the corresponding melody and lyrics in terms of quantity, thereby diminishing the overall generation quality.

**Pitch Initialization.** We evaluate these four methods on the pure melody continuation task. As shown in Table 5, the scalar initialization presents a considerable advantage over other methods. We conjecture that the scalar method provides a strong prior on both the magnitude and direction of the newly initialized embeddings, which induces the model to learn the pitch patterns comprehensively.

Table 5: Ablation study on pitch initialization methods.

| Init Method | Average | Gaussian | Interpolation | Scalar |
|---|---|---|---|---|
| MD ↓ | 3.07 | 3.41 | 2.9 | 2.33 |
| RR ↑ | 1.44 | 1.83 | 1.77 | 2.03 |

**Motif-level Melody Data.** To determine whether motif-level patterns enhance melody generation, we conduct baseline experiments where we train SongComposer exclusively on a pure melody dataset. We then test melody continuation, reporting melody distance (MD) and recall rate. We adjust the repeat threshold to control the repetition level of the motif-level data. The results are presented in Table 6, with the baseline result in the right-most column where no motif-level data is inputted.

Firstly, all results with motif-level data boost the baseline in terms of recall rate, aligning with our intuition that injecting motif-level data improves the structure awareness of melody composition. Secondly, we find that a small amount of highly repetitive motif-level data can hurt the melody generation. We conjecture this is because highly repetitive motifs lack diversity and trap the model in a constrained generation space. Then, the melody distance reaches an optimal point at a threshold of 10, suggesting that a moderate degree of repetition achieves the best balance between motif variety and overall repetitiveness. Therefore, we extract motif-level melodies by a repeat threshold of 10.

Table 6: Ablation on the repetition of motif-level melody data.

| Repeat threshold | 5 | 10 | 15 | 20 | 25 | ∞ |
|---|---|---|---|---|---|---|
| Quantity | 12.3× | 4.6× | 2.8× | 1.5× | 1× | 0× |
| RR ↑ | 2.19 | 2.03 | 1.61 | 1.58 | 1.51 | 1.45 |
| MD ↓ | 2.62 | 2.33 | 3.25 | 3.05 | 4.48 | 3.07 |

**Phrase-level Special Tokens.** To study the importance of phrase-level indication in song composition, we train the SongComposer model without phrase-level special tokens. The results, presented in Table 3, show a significant decline in generation quality when these tokens are omitted. Moreover, the use of phrase-level special tokens improves the model's ability to capture recurring musical ideas, as evidenced by an increase in the recall rate. Both observations suggest that phrase-level indications are essential for producing coherent and fluid song compositions.

To delve deeper into the influence of phrase tokens and the interplay between musical elements during generation, we categorize the input into four primary types: lyric, duration, pitch, and structure. The structure type here refers to phrase-level tokens. We then analyze the attention maps from all layers of SongComposer. The attention distribution shown in Figure 4 reveals that structural phrase-level tokens have a profound impact across all query types, underscoring the crucial role of structure in song generation. Furthermore, the model tends to prioritize musical elements that are consistent with the input query's type. For instance, when processing lyric queries, the model allocates nearly half of its attention to keys related to lyrics.

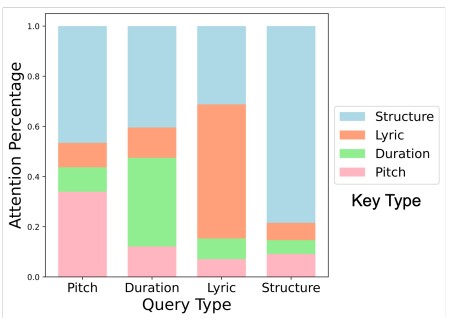

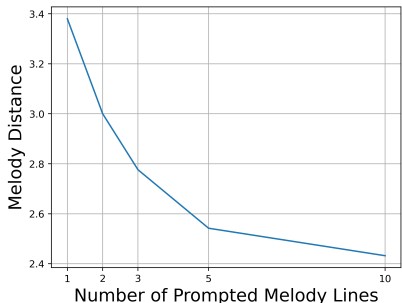

Figure 4: Visualization of attention distribution for different key/query types.

Figure 5: Memorization analysis of Song-Composer.

**Memorization analysis on SongComposer.** To investigate the extent to which SongComposer memorizes the training data, we conduct a memorization analysis inspired by MusicLM (Agostinelli et al., 2023). Specifically, we prompt training melody data samples with varying numbers of lines and compare the generated melodies to their original target counterparts. We quantify the similarity between the two melodies using melody distance, which would approach 0 if the prediction exactly matches the target. As shown in Figure 5, we find that the melody distance remains relatively high even when prompted with 10 lines of melody, indicating that our strategy is not trivially memorizing the training data and the generated results differ from the sequences in the training set.

## 5 CONCLUSION

In this paper, we introduce SongComposer, a novel large language model designed to generate detailed music scores that synchronize lyrics and melodies. The model leverages a tuple format to align lyrics and notes at the word level. Additionally, SongComposer employs scalar initialization for note pitch, which facilitates the efficient modeling of pitch information. When training on the large corps of song data, a multi-stage pipeline is implemented for structural capture, beginning with motif-level melody data and advancing to phrase-level indicators to enhance coherence and promote logical repetition. Our experiments show that SongComposer outperforms traditional methods and other large language models, including GPT-4, in tasks such as converting lyrics to melodies and vice versa, as well as in continuing existing songs and creating new songs from text. These results highlight SongComposer's potential as a valuable tool for assisting in music creation.

**Limitations and Future work.** SongComposer primarily focuses on generating symbolic music that synchronizes lyrics and melodies. However, producing corresponding audio currently requires supplementary singing voice synthesis tools. While the musical quality of the audio is partly dependent on the generated score (SongComposer's output), it also significantly relies on the singer's performance, including timbre and vocal techniques—areas outside the scope of symbolic music generation. This distinction is important for evaluating the performance of our model, as the perceived audio quality is heavily influenced by the synthesis tool used, not solely by our work. Additionally, SongComposer currently lacks the capability to generate multi-track accompaniments.

Symbolic music generation offers fine control and superior editability, whereas acoustic methods provide impressive musical expressiveness and listenability. In future work, we aim to integrate symbolic and acoustic approaches to create full-track songs. This integration will enable the generation of precise scores alongside their corresponding high-quality audio, achieving a balance between control and auditory appeal.

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

ETHICS STATEMENTS

The proposed work, SongComposer, a large language model designed for generating songs, has the potential impact on various aspects of society. On the positive side, SongComposer effortlessly creates high-quality songs with melodies and lyrics which can optimize the music creation process, allowing individuals with limited musical training to express their creativity and contribute to the music landscape.

However, as SongComposer generates songs autonomously, there is a risk of potential copyright infringement or misuse of intellectual property. We have conducted a preliminary memorization analysis shown in Figure 5. However, proper measures still need to be in place to ensure that the generated songs adhere to copyright laws and protect the rights of original composers and authors.

In conclusion, while SongComposer presents exciting possibilities for the music industry and creative expression, its development should be accompanied by careful consideration of ethical and societal implications.

## A    SONGCOMPOSE DATASET

This section introduces the compilation, creation, and statistical breakdown of our SongCompose dataset, which includes separate collections of lyrics, melodies, and lyric-melody pairs that synchronize lyrics with melodies at the word level. We aim to publicly release this three-fold dataset and following supervised fine-tuning dataset, providing a foundational resource for future research.

### A.1    PURE-LYRIC DATASET

We collect pure lyrics datasets from two online sources: (1) The Kaggle dataset[3], comprising the lyrics of 150K songs labeled with Spotify Valence, a measure of the positiveness of the song. (2) The Music Lyric Chatbot dataset[4], containing the lyrics of 140K Mandarin-language songs. After a series of lyric-cleaning processes, we gather high-quality lyrics from 283K songs, including 150K in English and 133K in Chinese.

Table 7 provides a detailed breakdown of the dataset, including language distribution, average lines per song, words per line, and the count of unique words.

| Language | #Song | #Line/#Song | #Word/#Line | #Unique word |
|---------|-------|-------------|-------------|--------------|
| English | 150359 | 34.4 | 5.8 | 168669 |
| Chinese | 132930 | 28.0 | 8.7 | 7740 |
| Total | 283289 | 31.4 | 7.0 | 176399 |

Table 7: Statistical details of the pure-lyric dataset.

### A.2    PURE-MELODY DATASET

To organize the melody dataset into a text-based structure, we collect MIDI files. Using MIDI files for our pure melody dataset offers inherent structural simplicity, enabling efficient extraction and manipulation of melodies without complex audio processing. Among our collection, 45K entries come from the LMD-matched MIDI dataset Raffel (2016), while approximately 80K are acquired through web crawling.

For parsing MIDI files, we employ `pretty_midi` Raffel & Ellis (2014), a Python module designed for creating, manipulating, and analyzing MIDI files. We extract the "melody" or "vocal" tracks from these MIDI files. Since melody in MIDI is represented as a sequence of musical notes over time and each note has a specific pitch, start and end timestamp, we obtain a list of melody attribute triplets consisting of {*note pitch*, *note duration*, *rest duration*}.

---

[3]https://www.kaggle.com/datasets/edenbd/150k-lyrics-labeled-with-spotify-valence
[4]https://github.com/liuhuanyong/MusicLyricChatbot

- *Note pitch:* The pitch of notes is represented by their corresponding MIDI note numbers, ranging from 0 to 127, with the number 60 predefined as Middle C.
- *Note duration:* A note's duration is defined as the length of time in seconds that the note is played. This is computed from the start and end times of each note embedded within the MIDI files as follows: note-duration$_k$ = note-end$_k$ − note-start$_k$, where $k$ represents the note index number.
- *Rest duration:* The rest duration represents the silent period that follows the playing of a note. It can be calculated by rest-duration$_k$ = note-start$_{k+1}$ − note-end$_k$.

We perform necessary data filtering to remove duplicate and poor-quality samples, leaving approximately 20K MIDI samples remaining.

## A.3    PAIRED LYRIC-MELODY DATASET

To build paired data with precise alignment, we process web-scraped information on a large scale efficiently, creating a dataset of 4K classic Chinese songs and 4K English songs. As illustrated in Figure 6, the pipeline for collecting lyric-melody data is as follows:

(1) Source Data Crawling: We crawl the web to gather a large dataset of mp3 files and their corresponding lyric files, encompassing sentence-level timestamps.

(2) Lyrics Cleaning: We use GPT-4 to clean irrelevant details from lyric texts, such as song titles, artist names, and production information.

(3) Segment Slicing: To mitigate the challenges and error accumulation for long-time alignments, we slice the audio and lyrics into paired segments of approximately 10 seconds (roughly three sentences each) based on timestamps provided in the lyric files.

(4) Music Source Separation: We utilize UVR[5], a public music separation tool, to separate the vocal from the accompaniment part in the original audio.

(5) Singing Voice Transcription: Using a singing voice wav input, FL Studio[6], a digital audio workstation software, automatically generates the preliminary musical score, capturing note pitch and start-end times of each note.

(6) Word Boundary Annotation: We obtain the boundaries of each word in lyrics with an audio alignment tool, Montreal Forced Aligner[7].

(7) Word-level Alignment: The dynamic time warping (DTW) Müller (2007) algorithm is utilized to align words and notes based on start-end times.

For information at the phrase level, we use the All-In-One music structure analyzer Kim & Nam (2023) to extract it. Finally, we develop a dataset comprising 8K paired lyric-melody entries, with approximately 4K in Chinese and 4K in English.

We also conduct the statistical analysis of the paired lyric-melody dataset shown in Figure 7. We find that most pitch numbers fall within the range of 50 to 80 and the majority of words are paired with a single note, and around 10% of words correspond to two or more notes. When examining note durations, we observe that they primarily vary between 0 to 1 second, and durations of rests are predominantly zero, reflecting a concise musical structure.

## A.4    SUPERVISED FINETUNING DATA

To achieve the instruction-following capability, we create supervised fine-tuning data for Song-Composer. For lyric-to-melody, melody-to-lyric, and song continuation tasks, we manually design the prompt templates in Figure 8, which serve as the foundation for compiling our QA pairs. For example, in the lyric-to-melody task, we start with the instruction prompt, such as "Please generate an appropriate melody for the provided lyrics." Then the pure-lyric version of a song follows the prompt. The response then utilizes the lyric-melody paired version of the song. For the song continuation task, we will additionally specify the number of lines by which we want the model to extend the song. To create the dataset for the final text-to-song task, we leverage the GPT-4 API. We feed the paired

---

[5]`https://github.com/Anjok07/ultimatevocalremovergui`
[6]`https://www.image-line.com/fl-studio`
[7]`https://github.com/MontrealCorpusTools/Montreal-Forced-Aligner`

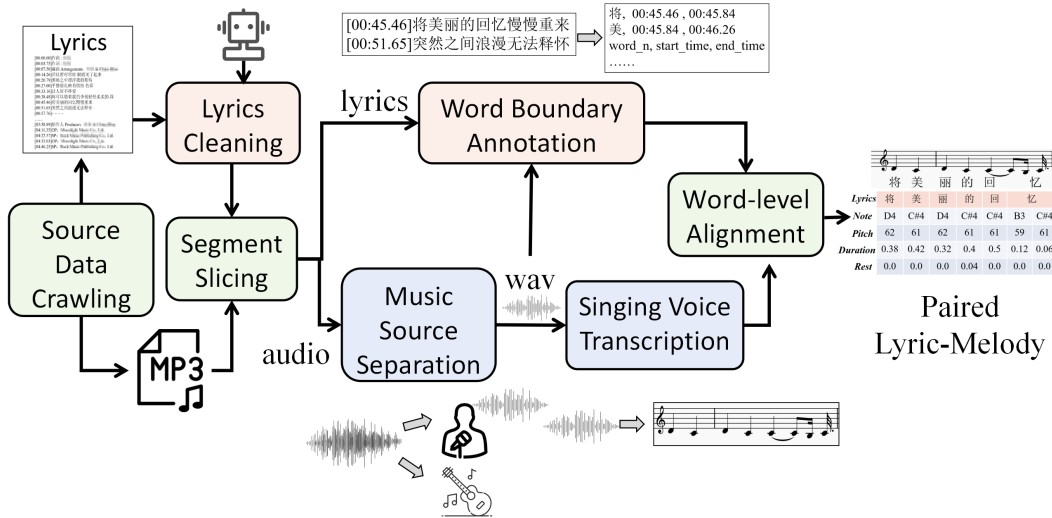

Figure 6: Pipeline of paired lyric-melody data collection.

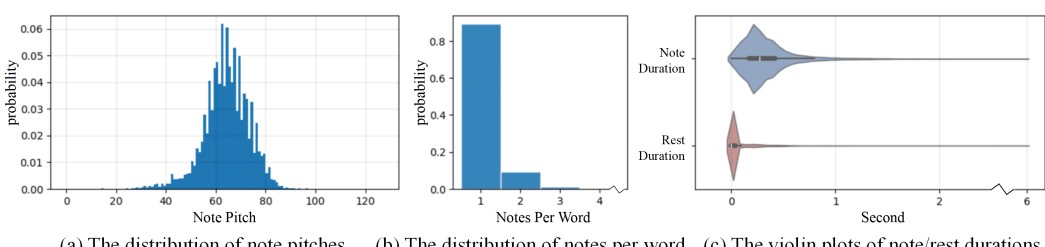

(a) The distribution of note pitches.    (b) The distribution of notes per word.    (c) The violin plots of note/rest durations.

Figure 7: Distribution of music attributes in our paired lyric-melody dataset.

song data into the model and ask it to generate a prompt. We use a few-shot template to guide the output, as shown in Figure 9. Therefore, we can compile the text-to-song instruction set.

## B  DETAILS ON SONG-RELATED GENERATION TASKS AND SUBJECTIVE METRICS

**Lyric-to-Melody Generation** asks to create a fitting melody based on the given lyrics. The melody is assessed on: (1) Harmony (HMY.): Evaluates the overall quality of the melody. (2) Melody-Lyric Compatibility (MLC.): Examines how well the generated melody fits the given lyrics.

**Melody-to-Lyric Generation** aims to produce lyrics that match a provided melody. The lyrics are evaluated on: (1) Fluency (FLN.): Considers the grammatical correctness and semantic coherence of the generated lyrics. (2) Melody-Lyric Compatibility (MLC.): Examines how well the generated lyrics fit the given melody.

**Song Continuation** involves extending a given song segment both melodically and lyrically. We evaluate the continuation quality on: (1) Overall Quality (OVL.): Measures the overall quality of the generated song in terms of its musical appeal. (2) Coherence to the Song Prompt (COH.): Analyzes the natural integration of the continuation with the provided song prompt, assessing coherence in melody, lyrics, and other musical elements.

**Text-to-Song Generation** generates a complete song based on textual description, capturing its essence musically and lyrically. The evaluation focuses on: (1) Overall Quality (OVL.): Measures the overall quality of the generated song in terms of its musical appeal. (2) Relevance to the Text Input (REL.): Examines how well the generated song aligns with and derives relevance from the input text.

> **Lyric-to-Melody:**
> - "Please generate an appropriate melody for the provided lyrics."
> - "Given the following lyrics, create a suitable melody."
> - "Craft a melody that complements these lyrics."
> - "Compose a tune in harmony with the accompanying lyrics."
> - "Generate a melody line that matches the given lyrics."
>
> **Melody-to-Lyric:**
> - "Compose a set of lyrics that align with the provided melody."
> - "Given the following melody, create corresponding lyrics."
> - "Write lyrics that harmonize with this melody."
> - "Create lyrics to accompany the given melody."
> - "Construct a lyric sequence that matches the provided melody."
>
> **Song Continuation:**
> - "Based on the existing song script, please write an additional {} lines for the song."
> - "Using the existing song script as a basis, please compose an additional {} lines for the song."
> - "To further develop the current song script, write {} more lines."
> - "Please expand upon the present song script by crafting an extra {} lines."
> - "Continue the existing song script by adding {} additional lines."

Figure 8: Instruction prompt templates for lyric-to-melody, melody-to-lyric, and song continuation tasks.

> ```python
> messages =[ ["role":"system", "content": f""" You are a helpful and precise assistant for
> constructing question-answer pairs. The song is represented in a three-part set format, each
> consisting of a musical note, the note's duration and the corresponding lyrics. Duration is
> represented with unit of sixteenth note, with greater values indicating longer durations, the
> maximum is 256. You will be provided with a song to write a one-sentence prompt and
> make sure to use simple words and be abstract. For example, 1. Craft a slow-paced English
> song for the holidays. 2. Compose a Mandarin song about friendship. 3. Make a English
> song about moonlight. 4. Make a happy English song. 5. Write a English song about the
> simple joys of a sunny day. 6. Pen a Chinese tune about finding love in unexpected places. 7.
> Compose a heartfelt English song about the warmth of home. 8. Create a English song
> about the adventure of a road trip. 9. Write a Chinese melody celebrating the feeling of
> freedom. 10. Craft a song about the beauty of the changing seasons. 11. Develop a English
> tune about the excitement of a first date. 12. Make a English song about the peace found in
> nature. 13. Write a song about overcoming challenges. 14. Compose a Chinese song that
> expresses gratitude for life's blessings. 15. Create a Mandarin song about the magic of a
> starry night. """}]
>
> for sample in fewshot_samples:
>     messages.append({"role":"user", "content":sample['song']})
>     messages.append(("role";"assistant", "content":sample['instruction']})
> ```

Figure 9: The prompt on text-to-song dataset construction process for GPT-4, using few-show in-context learning instructions.

| Pure-lyric | Pure-melody | MD $\downarrow$ | BS $\uparrow$ |
|:---:|:---:|:---:|:---:|
| ✗ | ✗ | 2.59 | 0.603 |
| ✗ | ✓ | 2.29 | 0.618 |
| ✓ | ✗ | 2.44 | 0.645 |
| ✓ | ✓ | 2.12 | 0.662 |

Table 8: Ablation study on pretraining datasets. ✗ denotes the exclusion of a specific dataset, while ✓ indicates its inclusion in the training. Paired data are used in all settings.

## C    MORE INFORMATION ON ABLATION STUDY

### C.1    EXAMPLE ON SONG-LEVEL AND LINE-LEVEL PAIR FORMAT

In practice, the input of the song-level paired melody is formatted as follows:

$\langle$bop$\rangle\langle$bol$\rangle$ Chinese/English song. Total $\{num\}$ lines.

The 1-st line: $w_1\,w_2\,\cdots$

The 2-nd line: $\cdots\,\langle$eol$\rangle$

$\langle$bom$\rangle$ bpm is $\{bpm\}$. Total $\{num\}$ lines.

The 1-st line: $\langle p_1\rangle, d_1\,|\,\langle$rest$\rangle, r_1\,|\,\langle p_2\rangle, d_2\,|\,\langle$rest$\rangle, r_2\cdots$

The 2-nd line: $\cdots\,\langle$eom$\rangle\langle$eop$\rangle$

The input of the line-level paired melody is formatted as follows:

$\langle$bop$\rangle$ Chinese/English song.  bpm is $\{bpm\}$. Total $\{num\}$ lines.

The 1-st line: $\langle p_1\rangle, d_1\,|\,\langle$rest$\rangle, r_1\,|\,\langle p_2\rangle, d_2\,|\,\langle$rest$\rangle, r_2\cdots||w_1\,w_2\,\cdots$

The 2-nd line: $\cdots\,\langle$eop$\rangle$

### C.2    ABLATION ON INDEPENDENT LYRIC AND MELODY TRAINING

To explore the impact of specialized datasets on our model's learning, we conduct training experiments using paired data combined with different pure-lyric and pure-melody datasets. Table 8 demonstrates that omitting both the pure-lyric and pure-melody datasets significantly reduces performance, highlighting the critical importance of foundational melodic and lyrical knowledge in the training stages.

Integrating either dataset individually results in notable improvements across tasks. Specifically, the pure-lyric dataset mainly enhances performance in the lyric-related generation, while the pure-melody dataset significantly boosts melody generation. This finding aligns with the intuitive understanding that each dataset enhances the model's comprehension of its respective modality. Moreover, using both types of datasets together yields the best results, demonstrating a synergistic effect.

## D    TUPLE FORMAT EXAMPLES

During the pretraining stage, we introduce three types of data. We give examples of what each format looks like. As illustrated in Figure 10, we present Chinese and English instances of pure lyrics. The structure for pure melody is exemplified in Figure 11. For lyric-melody pairs, bilingual versions are showcased in Figure 12.

## E    BASELINE CONSTRUCTION

### E.1    GPT

We invoke the GPT API to retrieve baseline results. We utilize a few-shot prompt to offer a template and instruct the model to follow suit. The pseudocode is illustrated in Figure 13.

**English:** "<bol> English song. Total 17 lines.\n<boverse> The 1-th line:well angel morning sivanna\n The 2-th line:well ain't been gone too far\n The 3-th line:but heading out towards ponoma \n<eoverse>\n<bochorus> The 4-th line:where you won't be alone\n The 5-th line:where there's'is thrift store manager in a poke camadee \n The 6-th line:and a gas mask on his arm\n<eochorus>\n<boother> The 7-th line:brick layer\n<eoother>\n<bochorus> The 8-th line:with a hat down on his feet\n The 9-th line:i'll say no more\n The 10-th line:i won't lead no calvary \n The 11-th line:how long\n The 12-th line:will you disregard the heat\n The 13-th line:half beat\n The 14-th line:it's no mismonor though\n The 15-th line:i've the feeling that i better go\n The 16-th line:so\n<eochorus>\n<booutro> The 17-th line:i slide right out the door oh\n<eooutro>\n<eol>"

**Chinese:** "<bol> Chinese song. Total 28 lines.\n<boverse> The 1-th line:北风在吹着清冷的街道\n The 2-th line:街灯在拉开长长的影子\n The 3-th line:走过的路想过的事\n The 4-th line:仿佛越来越远越来越长\n The 5-th line:越来越多越难以抛开\n<eoverse>\n<boverse> The 6-th line:多少平淡日子以来的夜晚\n The 7-th line:你曾是我渴望拥有的期盼\n<eoverse>\n<boverse> The 8-th line:没有人能挽回时间的狂流\n The 9-th line:没有人能誓言相许永不分离\n<eoverse>\n<bochorus> The 10-th line:没有人能挽回时间的狂流\n The 11-th line:没有人能了解聚散之间的定义\n The 12-th line:太多遗憾太多伤感\n<eochorus>\n<bochorus> The 13-th line:留在心中像一道狂流\n<eochorus>\n<boverse> The 14-th line:没有人能挽回时间的狂流\n The 15-th line:没有人能誓言相许永不分离\n The 16-th line:是我的错是你错过\n The 17-th line:喔\n<eoverse>\n<bochorus> The 18-th line:没有人能挽回时间的狂流\n The 19-th line:没有人能了解聚散之间的定义\n The 20-th line:太多遗憾太多伤感\n<eochorus>\n<boverse> The 21-th line:留在心中像一道狂流\n<eoverse>\n<boverse> The 22-th line:多少平淡日子以来的夜晚\n The 23-th line:你曾是我渴望拥有的企盼\n The 24-th line:没有人\n<eoverse>\n<bochorus> The 25-th line:没有人\n The 26-th line:没有人能了解\n The 27-th line:没有人能了解\n The 28-th line:没有人\n<eochorus>\n<eol>"

Figure 10: Two examples of lyric data in English and Chinese with phrase-level tokens.

" <bom> bpm is 143. Total 15 lines.\n<boverse> The 1-th line:<70>, 14 | <rest>, 12 | <67>, 2 | <rest>, 16 | <67>, 2 | <70>, 31 | <72>, 20 | <67>, 30 | <65>, 18 | <rest>, 14 | <68>, 28 | <67>, 25 | <65>, 9\n The 2-th line:<65>, 38 | <63>, 5 | <rest>, 48 | <68>, 3 | <67>, 2 | <68>, 6 | <67>, 3 | <68>, 8 | <68>, 18 | <66>, 4 | <66>, 9 | <61>, 11 | <rest>, 40 | <61>, 20\n The 3-th line:<64>, 25 | <61>, 46 | <rest>, 38\n<eoverse>\n<bochorus> The 4-th line:<69>, 16 | <68>, 1 | <67>, 11 | <rest>, 67 | <68>, 9 | <68>, 9 | <68>, 9 | <70>, 16 | <67>, 22\n The 5-th line:<68>, 3 | <rest>, 54 | <74>, 24 | <75>, 11 | <75>, 11 | <71>, 27 | <72>, 19 | <67>, 35 | <68>, 13 | <rest>, 49 | <65>, 20\n The 6-th line:<74>, 3 | <79>, 10 | <75>, 6\n<eochorus>\n<bochorus> The 7-th line:<77>, 13 | <70>, 17 | <67>, 27 | <68>, 12 | <rest>, 79 | <70>, 76 | <rest>, 256 | <70>, 22\n <eochorus>\n <boverse> The 8-th line:<70>, 8 | <68>, 21 | <rest>, 11 | <70>, 22 | <rest>, 16 | <68>, 5 | <rest>, 82 | <70>, 21 | <67>, 11 | <65>, 34 | <rest>, 232\n<eoverse>\n<boverse> The 9-th line:<70>, 11 | <rest>, 12 | <67>, 19 | <67>, 4 | <70>, 33 | <72>, 30 | <67>, 28 | <65>, 21 | <rest>, 9 | <68>, 32 | <67>, 23 | <65>, 8 | <65>, 47 | <rest>, 45\n The 10-th line:<67>, 16 | <68>, 5 | <68>, 5 | <rest>, 108\n<eoverse>\n<bochorus> The 11-th line:<61>, 1 | <61>, 1 | <rest>, 72\n The 12-th line:<68>, 22 | <67>, 1 | <67>, 1 | <rest>, 69 | <67>, 7 | <68>, 6 | <68>, 3 | <67>, 3 | <69>, 7 | <70>, 16 | <67>, 3 | <68>, 3\n The 13-th line:<67>, 21 | <rest>, 51 | <74>, 22 | <75>, 19 | <71>, 25 | <72>, 19 | <67>, 22 | <rest>, 92\n The 14-th line:<67>, 34 | <70>, 25 | <65>, 25 | <67>, 35 | <rest>, 59\n<eochorus>\n<bochorus> The 15-th line:<75>, 12 | <75>, 28 | <70>, 16 | <68>, 6 | <68>, 6 | <68>, 6 | <68>, 6 | <68>, 6 | <rest>, 256\n<eochorus>\n<eom>"

Figure 11: An example of pure melody data with phrase-level tokens.

## E.2 OPEN SOURCE LLM

For the open-source LLM, we select the base model as a fair comparison for all candidates. The prompt for the LLM is structured as follows:

$$\text{"system messages: } Q1 \rightarrow A1, Q2 \rightarrow A2, \ Q3 \rightarrow \text{"}$$

where system messages are the same as the one for GPT, $Q1$, $Q2$, and $A1$, $A2$ are examples of the tasks we want the model to perform. We instruct the model to generate $A3$ as the continuation of this prompt.

## F CASE STUDY: EVALUATING WELL-STRUCTURED SONG GENERATION

To better validate SongComposer's ability to generate well-structured songs, we conducted a case study. Figures 14 and 15 present examples of text-to-song generation in Chinese and English, respectively. We used different colored boxes to highlight phrase-level repetitions, different colored circles to mark motif-level repetitions, and underlines to indicate lyrical repetitions.

**English:** " \<bop\> English song. bpm is 94. Total 16 lines.\n\<boverse\> The 1-th line:\<70\>,6,and | \<69\>,8,you | \<67\>,9,don't | \<69\>,8,seem | \<70\>,7,to | \<72\>,12,to1 | \<74\>,10,to2 | \<67\>,30,understand\n The 2-th line:\<69\>,3,a | \<65\>,18,a1 | \<rest\>,14 | \<69\>,15,shame | \<67\>,8,you | \<69\>,9,seemed | \<70\>,8,an | \<72\>,12,honest | \<72\>,22,honest1 | \<70\>,18,man | \<69\>,19,man1 | \<rest\>,15\n The 3-th line:\<70\>,8,and | \<69\>,8,all | \<67\>,7,the | \<69\>,9,fears | \<70\>,8,you | \<72\>,12,hold | \<74\>,11,so | \<67\>,28,dear | \<69\>,4,dear1\n The 4-th line:\<65\>,17,will | \<rest\>,14 | \<70\>,5,will1 | \<69\>,9,turn | \<67\>,9,to | \<69\>,6,whisper | \<70\>,9,whisper1 | \<72\>,12,in | \<72\>,11,your | \<72\>,11,your1 | \<70\>,19,ear | \<69\>,8,ear1 | \<rest\>,25\n The 5-th line:\<67\>,2,and | \<67\>,2,you | \<74\>,7,know | \<74\>,6,what | \<72\>,6,they | \<74\>,6,say | \<72\>,4,might | \<74\>,13,hurt | \<72\>,10,you\n The 6-th line:\<67\>,2,and | \<67\>,4,you | \<74\>,4,know | \<74\>,2,that | \<74\>,4,it | \<74\>,7,means | \<72\>,3,so | \<74\>,6,much | \<72\>,13,much1 | \<rest\>,11\n The 7-th line:\<70\>,7,and | \<69\>,8,you | \<67\>,7,don't | \<69\>,8,even | \<70\>,8,even1 | \<72\>,11,feel | \<67\>,11,a | \<72\>,13,a1 | \<70\>,19,thing | \<69\>,24,thing1 | \<rest\>,10\n\<eoverse\>\n\<bochorus\> The 8-th line:\<79\>,14,i | \<77\>,15,am | \<72\>,9,falling | \<72\>,9,i | \<74\>,19,am | \<79\>,7,fading | \<77\>,16,fading1\n The 9-th line:\<72\>,17,i | \<74\>,17,have | \<79\>,7,lost | \<77\>,10,it | \<72\>,23,it1 | \<72\>,13,all | \<74\>,7,all1 | \<70\>,11,all2 | \<72\>,33,all3 | \<rest\>,18\n The 10-th line:\<77\>,99,and | \<74\>,6,you | \<77\>,6,you1 | \<70\>,14,you2 | \<67\>,16,don't | \<rest\>,133 | \<77\>,7,seem | \<74\>,5,the | \<77\>,19,the1 | \<74\>,5,the2 | \<77\>,7,lying | \<74\>,5,kind | \<77\>,6,kind1 | \<74\>,4,kind2 | \<77\>,22,kind3 | \<74\>,16,kind4 | \<77\>,20,kind5 | \<74\>,8,kind6 | \<72\>,3,kind7 | \<70\>,6,kind8 | \<67\>,6,kind9 | \<rest\>,62\n The 11-th line:\<79\>,8,i | \<77\>,10,i1 | \<72\>,5,am | \<72\>,25,falling | \<75\>,1,i | \<75\>,1,am | \<75\>,1,fading | \<rest\>,11\n The 12-th line:\<79\>,6,i | \<77\>,12,am | \<72\>,22,drowning | \<72\>,10,drowning1 | \<rest\>,13\n The 13-th line:\<79\>,6,help | \<77\>,14,me | \<72\>,21,to | \<74\>,15,breathe | \<70\>,8,breathe1 | \<70\>,7,breathe2 | \<70\>,7,breathe3 | \<74\>,1,breathe4 | \<75\>,6,breathe5 | \<77\>,7,breathe6 | \<74\>,17,breathe7 | \<79\>,5,breathe8\n The 14-th line:\<79\>,14,i | \<77\>,14,am | \<72\>,20,hurting | \<74\>,18,hurting1 | \<79\>,7,i | \<77\>,10,have | \<72\>,19,lost | \<72\>,8,it | \<72\>,10,all\n The 15-th line:\<79\>,6,i | \<77\>,13,i1 | \<72\>,2,am | \<72\>,17,losing\n\<eochorus\>\n\<booutro\> The 16-th line:\<74\>,17,help | \<70\>,8,me | \<70\>,7,me1 | \<70\>,7,to | \<77\>,15,to1 | \<74\>,44,breathe | \<72\>,11,breathe1 | \<rest\>,27 | \<72\>,3,breathe2 | \<rest\>,256\n\<eooutro\>\n\<eop\>"

**Chinese:** " \<bop\> Chinese song. bpm is 113. Total 12 lines.\n\<boverse\> The 1-th line:\<70\>,27,风 | \<70\>,12,烟 | \<67\>,19,烟1 | \<75\>,20,滚 | \<70\>,30,滚1 | \<72\>,9,滚 | \<67\>,9,滚1 | \<65\>,9,唱 | \<63\>,11,唱1 | \<rest\>,13 | \<65\>,19,唱2 | \<67\>,7,唱3 | \<70\>,9,英 | \<65\>,9,英1 | \<63\>,7,英2 | \<60\>,20,英3 | \<58\>,26,雄\n\<eoverse\>\n\<boverse\> The 2-th line:\<63\>,26,四 | \<65\>,12,面 | \<67\>,18,面1 | \<70\>,20,面2 | \<72\>,8,面3 | \<70\>,8,面4 | \<67\>,20,面5 | \<70\>,24,面6 | \<rest\>,19 | \<70\>,27,面7 | \<67\>,9,面8 | \<65\>,19,面9 | \<63\>,7,面10 | \<65\>,4,青 | \<65\>,4,山 | \<65\>,4,侧 | \<67\>,20,耳 | \<67\>,20,听 | \<67\>,20,侧 | \<rest\>,24 | \<67\>,1,耳 | \<67\>,1,听 | \<67\>,1,；\n The 3-th line:\<67\>,27,晴 | \<70\>,9,天 | \<60\>,9,响 | \<60\>,9,雷 | \<63\>,41,敲 | \<70\>,19,金 | \<67\>,21,鼓 | \<rest\>,17\n The 4-th line:\<67\>,13,大 | \<70\>,10,海 | \<72\>,18,扬 | \<77\>,20,波 | \<77\>,18,作 | \<74\>,9,和 | \<72\>,12,和1 | \<70\>,39,声 | \<rest\>,26\n The 5-th line:\<75\>,36,人 | \<74\>,18,民 | \<72\>,37,战 | \<67\>,28,士 | \<rest\>,12 | \<70\>,38,驱 | \<72\>,14,虎 | \<70\>,7,豹 | \<72\>,9,豹1\n\<eoverse\>\n\<boverse\> The 6-th line:\<72\>,17,舍 | \<75\>,14,生 | \<77\>,22,忘 | \<79\>,51,忘1 | \<77\>,4,死 | \<75\>,12,死1 | \<67\>,53,保 | \<70\>,12,保1 | \<rest\>,8 | \<72\>,16,保2 | \<77\>,17,保3 | \<75\>,8,保4 | \<77\>,8,和 | \<72\>,21,和1 | \<70\>,118,平 | \<rest\>,256\n\<eoverse\>\n\<boverse\> The 7-th line:\<74\>,16,英 | \<72\>,15,雄 | \<74\>,15,的 | \<76\>,3,的1 | \<77\>,26,鲜 | \<69\>,50,血 | \<72\>,10,染 | \<76\>,21,红 | \<74\>,9,了 | \<72\>,61,它 | \<rest\>,17\n\<eoverse\>\n\<bochorus\> The 8-th line:\<72\>,13,为 | \<74\>,13,什 | \<77\>,8,么 | \<77\>,8,大 | \<79\>,45,地 | \<81\>,13,春 | \<77\>,14,春1 | \<76\>,15,春2 | \<74\>,15,常 | \<72\>,15,在 | \<74\>,62,在1 | \<rest\>,16\n\<bochorus\> The 9-th line:\<69\>,10,为 | \<72\>,12,什 | \<74\>,13,么 | \<77\>,37,么1 | \<74\>,7,战 | \<72\>,6,战1 | \<69\>,13,战2 | \<72\>,25,旗 | \<74\>,10,美 | \<62\>,24,美1 | \<65\>,19,如 | \<67\>,6,如1 | \<69\>,51,画 | \<rest\>,15\n\<eochorus\>\n\<bochorus\> The 10-th line:\<74\>,9,英 | \<72\>,12,雄 | \<74\>,12,的 | \<77\>,24,鲜 | \<69\>,25,血 | \<69\>,17,染 | \<72\>,7,红 | \<76\>,17,红1 | \<74\>,8,了 | \<72\>,52,它 | \<rest\>,16\n The 11-th line:\<72\>,11,为 | \<74\>,13,什 | \<77\>,13,麼 | \<79\>,37,大 | \<81\>,12,地 | \<77\>,16,春 | \<76\>,9,春1 | \<74\>,12,常 | \<72\>,14,常1 | \<74\>,48,在 | \<rest\>,15\n The 12-th line:\<74\>,12,英 | \<77\>,12,雄 | \<79\>,13,的 | \<81\>,50,生 | \<76\>,36,命 | \<74\>,10,命1 | \<rest\>,8 | \<72\>,12,命2 | \<69\>,6,命3 | \<72\>,6,开 | \<74\>,12,鲜 | \<77\>,13,鲜1 | \<79\>,17,鲜2 | \<77\>,6,鲜3 | \<79\>,10,鲜4 | \<81\>,103,花 | \<rest\>,13 | \<77\>,209,花1 | \<rest\>,54\n\<eochorus\>\n\<eop\> "

Figure 12: Two examples of lyric-melody pair data in English and Chinese with phrase-level tokens.

```python
messages = [ {"role":"system", "content": f""" You serve as an efficient and meticulous
assistant in addressing the task of lyric and melody composition in song generation. The
final output is represented in a three-part set format, each consisting of a musical note,
the note's duration and the corresponding lyrics. Duration is represented with unit of
sixteenth note, with greater values indicating longer durations, the maximum is 256.
Higher numbers signify extended durations. Please diligently follow this convention
when providing your response and strictly align the pattern with the sample provided
below. """} ]
for sample in fewshot_samples:
    messages.append({"role":"user", "content":sample['question']})
    messages.append({"role";"assistant", "content":sample['answer']})
```

Figure 13: The prompt construction process for GPT-3.5/GPT-4, using few-shot in-context learning instructions.

In these cases, we can observe distinct differences in SongComposer's handling of verses and choruses. Figure 14 clearly exhibits phrase-level repetitions, while Figure 15 demonstrates significant motifs. Notably, our lyrics harmonize with the melody, particularly in segments where the melody repeats, demonstrating semantic alignment.

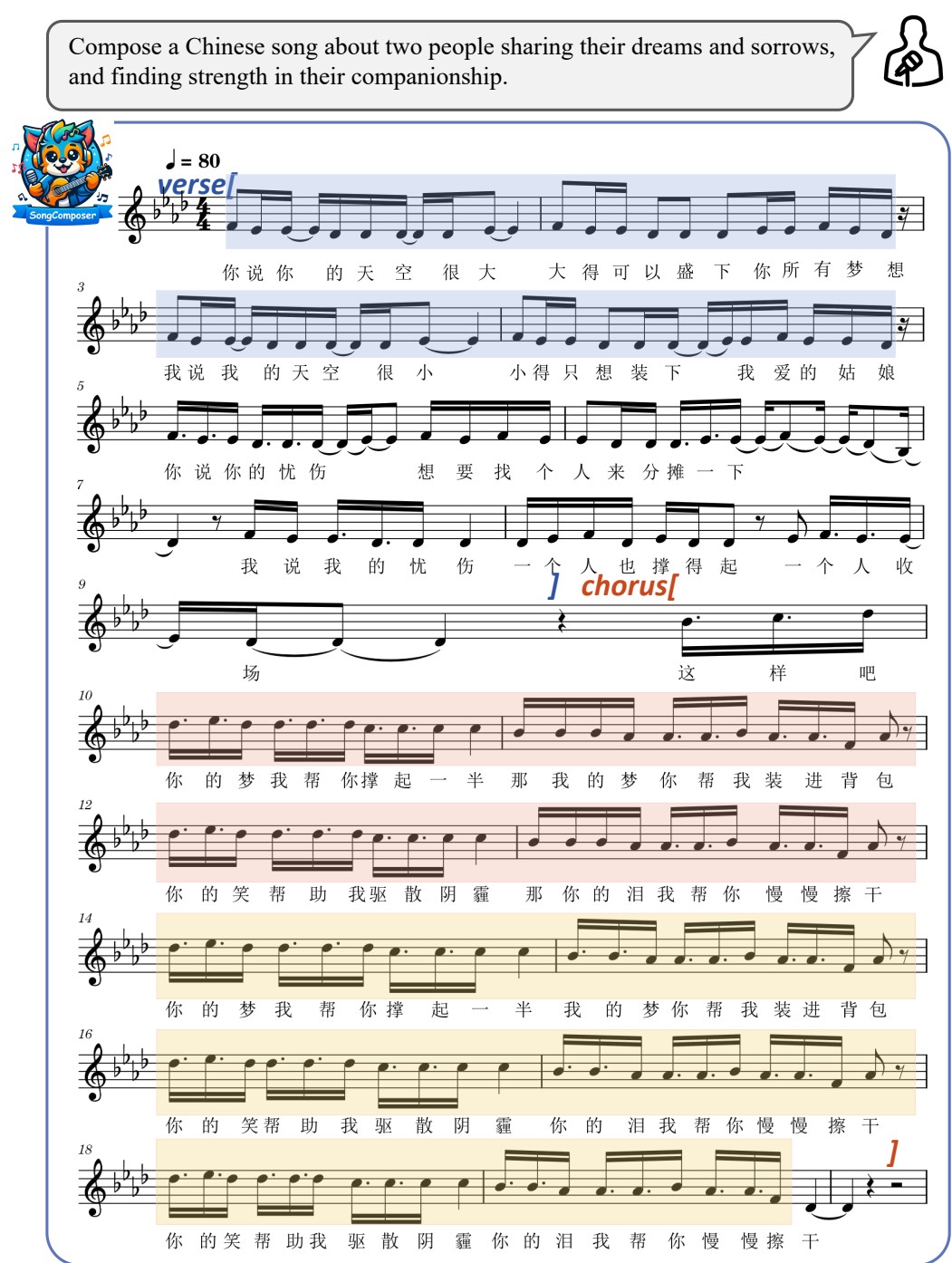

Figure 14: A text-to-song example in Chinese, featuring clear phrase-level repetitions highlighted with different colored boxes.

Figure 15: A text-to-song example in English, featuring prominent motif-level repetitions marked with different colored circles.

