# OpenReview forum: "SongComposer: A Large Language Model for Lyric and Melody Composition in Song Generation"
_ICLR.cc/2025/Conference — Submitted to ICLR 2025_

### Official Review · Reviewer_N9vU · 2024-10-21

**Soundness:** 3
**Presentation:** 3
**Contribution:** 2
**Rating:** 5
**Confidence:** 4

**Summary:**

The paper proposes SongComposer for multiple tasks including cross-generation between lyrics and melodies, song continuation, and text-to-song. SongComposer is based on InternLM2-7B and is trained by pure melody, pure lyrics, paired data, and instruction data, where different representations are designed for lyrics, melody, and paired data. The experiments show that SongComposer output performs other LLMs in these music-related tasks. However, the novelty is still limited for there is little modification of the model.

**Strengths:**

1. This paper design involves multi-stage training for SongComposer, making it capable of performing different tasks.
2. Numerous comparison and ablation experiments have been carried out to prove that SongComposer outperforms other models.

**Weaknesses:**

1. The novelty is still limited because SongComposer is based on InternLM2-7B with little modification, most of the work is on data processing.
2. The experiment setting needs further discussion. Since other LLMs are not for melody or lyrics tasks, it is obvious that they will have poor performance compared with SongComposer.

**Questions:**

1. Is there any measure taken to reduce the error accumulation when establishing the paired lyrics-melody data, considering the process involved in many pre-trained models that may introduce bias in each step?
2. What is the detailed finetune process for InternLM 2 in Table 1?

---

### Official Review · Reviewer_oqtZ · 2024-11-01

**Soundness:** 2
**Presentation:** 3
**Contribution:** 2
**Rating:** 5
**Confidence:** 3

**Summary:**

The paper introduces SongComposer, a pioneering large language model (LLM) designed for unified lyric and melody composition in song generation. It integrates capabilities for composing lyrics and melodies simultaneously, addressing the complex task of aligning melody and lyrics in a symbolic format.

**Strengths:**

The contributions and advantages of this work include the introduction of a novel scalar initialization for note pitches, the integration of motif- and phrase-level knowledge to enhance the model's understanding of pitch attributes and song structure, and the curation of the SongCompose dataset, which includes lyrics, melodies, and aligned lyric-melody pairs in both Chinese and English. SongComposer demonstrates superior performance over traditional composition models and advanced LLMs like GPT-4 in various song-related generation tasks.

**Weaknesses:**

1. This paper makes extensive use of open-source Large Language Models (LLMs) to process the dataset. The hallucination effect of large language models can lead to errors, such as the omission/generation of blank content and incorrect content, resulting in a lower quality dataset. The paper does not provide an expanded analysis or explanation of how to address this issue.
2. The model introduced in this paper is based on the existing capabilities of InternLM2-7B, with integration and subsequent optimization, suggesting that the model may lack in terms of innovation.
3. There are flaws in the evaluation: (1) Despite the paper's emphasis on alignment issues, it does not include a specific evaluation targeting alignment. (2) The evaluation metrics chosen in this paper are insufficient, particularly the subjective evaluation metrics.

**Questions:**

1. This paper makes extensive use of open-source Large Language Models (LLMs) to process the dataset. The hallucination effect of large language models can lead to errors, such as the omission/generation of blank content and incorrect content, resulting in a lower quality dataset. The paper does not provide an expanded analysis or explanation of how to address this issue.
2. The model introduced in this paper is based on the existing capabilities of InternLM2-7B, with integration and subsequent optimization, suggesting that the model may lack in terms of innovation.
3. There are flaws in the evaluation: (1) Despite the paper's emphasis on alignment issues, it does not include a specific evaluation targeting alignment. (2) The evaluation metrics chosen in this paper are insufficient, particularly the subjective evaluation metrics.

---

### Official Review · Reviewer_AoU6 · 2024-11-02

**Soundness:** 3
**Presentation:** 3
**Contribution:** 2
**Rating:** 5
**Confidence:** 4

**Summary:**

1. This paper proposes an LLM-based system for song composing, including melody and lyrics generation. The authors continue pretrain the InternLM2-7B model with their collected SongCompose dataset and a supervised-finetuning dataset for instruction-following song generation.
2. The paper proposed a reasonable musical data format and an extra musical token embedding initialization method for better LLM fine-tuning, and then they employ a curriculum learning method to enable the LLM to progressively learn structured song composition.
3. The generated results beat specialist lyrics-to-melody and melody-to-lyrics models and advanced LLMs like GPT-4.

**Strengths:**

1. Using LLMs to generate melody and lyrics is a very natural approach, as both melodies and lyrics can be easily transformed into formats suitable for LLM processing.
2. The experiments are thorough, and the demo webpage showcases examples with well-controlled song themes and effective handling of verse-chorus structure.
3. The authors' exploration of pitch token embedding provides a novel perspective on the integration of music and LLMs.

**Weaknesses:**

1. The methodological innovation is limited; the paper primarily reorganizes the data and explores different initialization methods for additional embeddings. It offers little insight into out-of-domain token embedding initialization for other LLM fine-tuning processes.
2. The paper lacks a comparison with existing LLM-based melody generation models, such as ChatMusician[1], and recent melody-to-lyrics generation models, like LOAF-M2L[2].
3. The generated Chinese lyrics do not appear to be very rhymed, and it's a pity that no improvements are proposed to address rhyming, as this is something that existing LLMs cannot achieve in a zero-shot manner.

[1] Yuan, Ruibin, et al. "Chatmusician: Understanding and generating music intrinsically with llm." arXiv preprint arXiv:2402.16153 (2024).
[2] Ou, Longshen, Xichu Ma, and Ye Wang. "Loaf-m2l: Joint learning of wording and formatting for singable melody-to-lyric generation." arXiv preprint arXiv:2307.02146 (2023).

**Questions:**

1. The melody dataset seems rather limited. Why not incorporate a larger melody dataset in ABC notation for training, such as IrishMan [1] (200k Irish ABC tunes)?
2. In Section 3.1 Pure Melody Format, how do you determine each line of the melody? As far as I know, MIDI files do not provide line / phrase segmentation for melodies.
3. I am puzzled as to why the Interpolation Initialization and Scalar Initialization methods proposed in Section 3.2 perform better. In theory, most melodies come from tonal music, where seven of the twelve pitches (natural scale) and the other five have asymmetrical semantic roles. However, the mathematical approach to embedding initialization does not align with common musical understanding. Are there any relevant references for the embedding initialization approach proposed in this paper?
[1] Wu, Shangda, et al. "TunesFormer: Forming Irish Tunes with Control Codes by Bar Patching." arXiv preprint arXiv:2301.02884 (2023).

---

### Official Review · Reviewer_ANGg · 2024-11-04

**Soundness:** 2
**Presentation:** 2
**Contribution:** 2
**Rating:** 3
**Confidence:** 4

**Summary:**

The paper introduces SongComposer, a novel large language model (LLM) specifically designed for song composition tasks. SongComposer can generate symbolic lyrics and melodies in a unified, instruction-following format.

The model addresses the challenge of integrating lyric and melody generation by using a word-level tuple format that aligns melody attributes with lyrics and a multi-stage training pipeline that enhances structural understanding of music through motifs and phrase-level indicators.

To support the training of SongComposer, the authors curated a comprehensive SongCompose dataset, including pure lyrics, pure melodies, and aligned lyric-melody pairs in both English and Chinese. This dataset aims to boost future research in music generation.

**Strengths:**

Unified Song Composition Model: SongComposer effectively combines lyrics and melody generation in one model, outperforming advanced LLMs like GPT-4 in song-related tasks. This unified approach is a significant step forward in simplifying the song composition process using AI.

Innovative Structural Training: The model’s use of motif-level and phrase-level training techniques ensures that the generated songs have better structural coherence and musical quality, which is crucial for realistic and engaging music.

**Weaknesses:**

Deficient Musicality in Melody Generation: Despite the novelty of the approach, the musical quality of the generated melodies falls short compared to what current advanced models can achieve.
1. In terms of end-to-end music generation, models like Jukebox and Suno deliver significantly more musically appealing results.
2. Using a pipeline approach, where a top-performing symbolic music model (music transformer or more recent development of whole song generation using hierarchical diffusion models) first generates the melody followed by adding lyrics, would also yield better musicality and coherence. The comparatively lower quality of SongComposer's melodies makes it less competitive in the rapidly advancing field of Music AI.

**Questions:**

The abstract mentioned understanding sheet music. Sheet music understanding, in music information retrieval, means understanding and analyzing music based on the image format (in a narrow sense, OMR). I found nothing in the paper; did you just mean that you extracted helpful information from text-ish-based score? If so, where does the structure label come from -- hand labeling or automated labeling using existing MIR tools? Thanks.

---

### Official Review · Reviewer_ThQn · 2024-11-09

**Soundness:** 2
**Presentation:** 3
**Contribution:** 2
**Rating:** 5
**Confidence:** 4

**Summary:**

SongComposer is a language model designed for song composition, utilizing symbolic song representations to handle both melody and lyrics in a structured format. SongComposer employs a tuple format that includes pitch, duration, and rest duration, allowing for alignment between lyrics and melody attempting to improve token efficiency. It is trained on SongCompose-PT, a dataset comprising lyrics, melodies, and paired lyric-melodies in Chinese and English, with additional QA pairs to enhance instruction-following for various song-related tasks. Experiments are being conducted on tasks like lyric-to-melody generation, melody-to-lyric generation, song continuation, and text-to-song creation, comparing SongComposer to other models.

**Strengths:**

- The paper introduces both a novel model, SongComposer, and a comprehensive dataset, SongCompose, which are substantial contributions to the field of AI-driven music composition. The dataset includes a large collection of lyrics, melodies, and precisely aligned lyric-melody pairs. The data would greatly benefit the research community if being open-sourced.
- This work is pioneering as it presents the first large language model capable of processing lyrics and melodies at the same time. Previous approaches typically handled these elements separately or focused on converting one to the other.
- The paper effectively demonstrates how scalar initialization disentangles pitch embeddings, which is a novel contribution to the methodology of music generation models. The visualization part showcases that pitch embeddings are meaningfully organized, helping the model understand and generate musical pitches more effectively.

**Weaknesses:**

1. **Lack of Evaluation on Music Understanding**
    - The paper frequently claims that SongComposer "understands" sheet music and musical structures. For example, the abstract mentions: "Sheet music understanding, we designed a flexible tuple format to load lyric and note attributes, fostering word-level alignment between lyrics and melodies, and enabling SongComposer to generate lyrics with accompanying well-aligned melodies." However, the paper does not provide concrete evaluations to substantiate the model's music understanding capabilities. The evaluations focus solely on generation tasks, without assessing the model's comprehension of musical concepts or structures.
    - To support the claim of "understanding," throughout the paper (e.g. `sheet music understanding`), the authors should include evaluations on music understanding tasks. For instance, Yuan et al. in ChatMusician [https://arxiv.org/abs/2402.16153] propose a college-level music understanding benchmark to evaluate such capabilities. Incorporating similar evaluation settings would strengthen the paper's claims. Additionally, given the three-stage training strategy on data that differs significantly from the original text distribution, there is a concern about catastrophic forgetting, which could diminish the model's general understanding abilities.
2. **Interpretability of the Objective Metrics**
    - The paper uses Melody Distance (MD) as a key metric to evaluate the similarity between generated melodies and ground truth. However, the interpretability of the MD values is unclear. In the memorization analysis section, the paper states that a melody distance of approximately 2.4 indicates that the generated melody is "far away" from the training data. Yet, in Table 1, the lowest MD value is 2.2, which is not significantly different from 2.4. This raises questions about the usefulness of MD as an informative metric.
    - Moreover, in practical experience, the connection between lyrics and melody is often weak. Therefore, low MD values might not necessarily reflect better performance in aligning lyrics with melodies. To verify the effectiveness of MD as a metric, the authors should consider constructing a simple baseline: select random lyrics from the training set, retrieve the most similar lyrics, and use their corresponding melodies to compute MD. If this baseline MD is also between 2 and 3, similar to the values reported in the paper, it would suggest that the MD metric may not effectively measure the quality of the model's generation. If it is a large number, then it would raise question about the memorization effect of the model, and whether it is safe to be open-sourced.
    - Additionally, the large MD values observed for models like LLaMA 2 and InternLM 2 might be due to invalid output formats, as these models have not been fine-tuned on the specific notation and may not produce outputs that conform to the expected format. This could artificially inflate their MD values, making direct comparisons less informative. Therefore, the paper should provide a more detailed analysis and discuss the limitations of the MD metric, possibly including additional baseline comparisons and considering alternative evaluation metrics.
    - Not just MD, I believe the simple retrieval based baseline should be implemented to give a better understanding of all objective metrics, including PD, DD, MD, Cosine, ROUGE2, BS.
    - (Note: There is also a typo in Table 1 where "Cosine Dist." should be "Cosine Sim.")
3. **Comparison with Established Music Notations**
    - The paper introduces a custom notation format for encoding lyrics and melodies. However, established notation systems like ABC notation already provide capabilities to encode lyrics, comments, notes, and chords effectively. Comparing the proposed notation with ABC notation would offer insights into the advantages or limitations of the new format. Such a comparison could address aspects like expressiveness, ease of use, compatibility with existing tools, and potential for broader adoption. Without this comparison, it is unclear why a new notation was necessary and how it improves upon existing standards.
4. **Fairness of Baseline Comparisons**
    - The paper compares SongComposer with other large language models (LLMs) like GPT-4 and GPT-3.5 using few-shot prompting. However, these models have not been trained on the specific notation introduced in the paper. This makes direct comparisons potentially unfair and less informative, as the performance gap may be due to unfamiliarity with the notation rather than differences in underlying capabilities. The baselines might not adequately reflect the strengths of the competing models in the context of song composition tasks. The paper would benefit from fine-tuning baseline models on the notation or designing evaluation tasks where all models have equal footing.

**Questions:**

I have no further questions. Please address my concerns especially on the objective metrics part.

---

### Meta-Review · Area_Chair_pBYN · 2024-12-20

**Metareview:**

The paper presents a model and a dataset that employs an interleaved lyrics and melody format for training a large language model on both song elements at the same time. The generated songs have better alighment and structural coherence compared to GPT-4 in lyric-to-melody generation, melody-to-lyric generation, song continuation, and text-to-song creation tasks. Other innovations of the paper are song note tokenizing based on a musical prior and structural music generation using a multi-stage pipeline starting with motif-level melody and later including special phrase-level tokens.

**Additional Comments On Reviewer Discussion:**

The reviewers acknowledged the novelty of the approach, but also pointed out concerns in evaluation, such as lack of evaluation on music understanding tasks, interpretability and fairness of baseline comparisons since other LLMs are not for melody or lyrics tasks and more. Some reviewers specifically pointed out to laking comparisons to ChatMusician, and melody-to-lyrics generation such as LOAF-M2L.
Since the authors did not respond to these reviews, the initial rankings remain below acceptance threshold.

---

### Decision · Program_Chairs · 2025-01-22

Reject